

# Charge neutralization as the major factor for the assembly of nucleocapsid-like particles from C-terminal truncated hepatitis C virus core protein

Theo Luiz Ferraz de Souza[1,2,*], Sheila Maria Barbosa de Lima[3,*], Vanessa L. de Azevedo Braga[2,4], David S. Peabody[5], Davis Fernandes Ferreira[2,6], M. Lucia Bianconi[4], Andre Marco de Oliveira Gomes[2,4], Jerson Lima Silva[2,4] and Andréa Cheble de Oliveira[2,4]

[1] Faculdade de Farmácia, Universidade Federal do Rio de Janeiro, Rio de Janeiro, Brazil
[2] Instituto Nacional de Ciência e Tecnologia de Biologia Estrutural e Bioimagem, Universidade Federal do Rio de Janeiro, Rio de Janeiro, Brazil
[3] Bio-Manguinhos, Fundação Oswaldo Cruz, Rio de Janeiro, Brazil
[4] Programa de Biologia Estrutural, Instituto de Bioquímica Médica Leopoldo de Meis, Universidade Federal do Rio de Janeiro, Rio de Janeiro, Brazil
[5] Department of Molecular Genetics and Microbiology and Cancer Research and Treatment Center, University of New Mexico, Albuquerque, United States
[6] Instituto de Microbiologia Paulo de Góes, Universidade Federal do Rio de Janeiro, Rio de Janeiro, Brazil
[*] These authors contributed equally to this work.

Corresponding authors
Jerson Lima Silva,
jerson_silva@uol.com.br,
jerson@bioqmed.ufrj.br
Andréa Cheble de Oliveira,
cheble@bioqmed.ufrj.br

## ABSTRACT

**Background**. Hepatitis C virus (HCV) core protein, in addition to its structural role to form the nucleocapsid assembly, plays a critical role in HCV pathogenesis by interfering in several cellular processes, including microRNA and mRNA homeostasis. The C-terminal truncated HCV core protein (C124) is intrinsically unstructured in solution and is able to interact with unspecific nucleic acids, in the micromolar range, and to assemble into nucleocapsid-like particles (NLPs) *in vitro*. The specificity and propensity of C124 to the assembly and its implications on HCV pathogenesis are not well understood.

**Methods**. Spectroscopic techniques, transmission electron microscopy and calorimetry were used to better understand the propensity of C124 to fold or to multimerize into NLPs when subjected to different conditions or in the presence of unspecific nucleic acids of equivalent size to cellular microRNAs.

**Results**. The structural analysis indicated that C124 has low propensity to self-folding. On the other hand, for the first time, we show that C124, in the absence of nucleic acids, multimerizes into empty NLPs when subjected to a pH close to its isoelectric point (pH $\approx$ 12), indicating that assembly is mainly driven by charge neutralization. Isothermal calorimetry data showed that the assembly of NLPs promoted by nucleic acids is enthalpy driven. Additionally, data obtained from fluorescence correlation spectroscopy show that C124, in nanomolar range, was able to interact and to sequester a large number of short unspecific nucleic acids into NLPs.

**Discussion**. Together, our data showed that the charge neutralization is the major factor for the nucleocapsid-like particles assembly from C-terminal truncated HCV core protein. This finding suggests that HCV core protein may physically interact with

![PeerJ]

unspecific cellular polyanions, which may correspond to microRNAs and mRNAs in a host cell infected by HCV, triggering their confinement into infectious particles.

## INTRODUCTION

Hepatitis C Virus (HCV) is a member of the *Flaviviridae* family, genus *Hepacivirus*, which causes acute and chronic liver diseases. It is estimated that about 170–200 million people worldwide are infected with HCV (*Pol et al., 2012*). Chronic hepatitis C can lead to cirrhosis and subsequent complications such as hepatocellular carcinoma (HCC). Although in remarkable development, current therapies against HCV are still unsatisfactory and several researchers seek a better understanding of processes associated to the HCV infection and pathogenesis in order to favor the development of novel alternative treatments (*Vermehren & Sarrazin, 2011*; *Wendt et al., 2014*; *Douam, Ding & Ploss, 2016*).

HCV contains one copy of a positive single-stranded RNA genome which encodes a single polyprotein with approximately 3,000 amino acids. This polyprotein is processed by both cellular and viral proteases at the level of the endoplasmic reticulum (ER) into at least 10 mature proteins (*Grakoui et al., 1993*). The structural proteins include the core protein and the envelope glycoproteins E1 and E2. The nonstructural proteins NS3, NS4A, NS4B, NS5A and NS5B are components of the viral replicase complex, and p7 and NS2 proteins are dispensable for replication but necessary for the production of progeny viruses (*Moradpour, Penin & Rice, 2007*; *Jones et al., 2007*; *Gentzsch et al., 2013*).

HCV infection is the leading cause of hepatocellular carcinoma (HCC) worldwide. Previous studies have the involvement of the core protein, NS3 and NS5A in the development of cancer (*Kasprzak & Adamek, 2008*; *Tran, 2008*). HCV core protein is a multifunctional protein that is involved in many viral and cellular processes. In addition to its structural role that involves viral RNA package and protection, the core protein has been reported to interact with many cellular proteins, thereby impacting immune presentation, apoptosis, cell transformation, lipid metabolism, and transcription, all of which being suggested to be related to HCV pathogenesis (*Kasprzak & Adamek, 2008*; *Irshad & Dhar, 2006*; *Polyak et al., 2006*; *McGivern & Lemon, 2008*; *Tang & Grisé, 2009*). *Moriya et al. (1998)* demonstrated that expression of core protein in transgenic mice induces hepatocellular carcinoma (HCC) development, suggesting a critical role in oncogenesis (*Moriya et al., 1998*). HCV core protein also promotes alteration in mRNA and microRNA homeostasis in hepatocytes (*Peng et al., 2009*; *De Giorgi et al., 2009*; *Steuerwald et al., 2010*; *Kao, Yi & Huang, 2016*). Deregulation of microRNA homeostasis may be a key pathogenetic factor in viral hepatitis and hepatocellular cancer (*Ruan, Fang & Ouyang, 2009*; *Chen, 2009*). A clear understanding of the mechanisms involved in microRNA deregulation in the Hepatitis C may lead to the development of new diagnostic and therapeutic strategies. Additionally, the core protein has potent nucleic acid chaperoning activities (*Cristofari et al., 2004*; *Ivanyi-Nagy et al.,*

*2006*), which involves the three basic clusters of N-terminal (*Sharma et al., 2010*). Different truncated hepatitis C virus core proteins can induce proapoptotic and pronecrotic effects (*Yan et al., 2005*), which might play an important role in the pathogenesis of HCV persistent infection and HCC.

The first cleavage of the polyprotein forms an immature core protein of 191 amino acids residues (*Lauer & Walker, 2001*). During the following cleavage, the core protein remains anchored to the ER through a C-terminal hydrophobic region (*Grakoui et al., 1993*) being further processed at its C-terminus by a signal peptide peptidase resulting in the mature protein (*Liu et al., 1997*; *Yasui et al., 1998*). The C-terminus of mature core protein is not precisely elucidated but probably lies between residues 170 and 182 (*Kopp et al., 2010*; *Oehler et al., 2012*). Although core protein has been shown to possess a nuclear localization signal (*Suzuki et al., 1995*; *Yan et al., 2005*), it is located exclusively in the cytoplasm in cells infected with cell culture-derived HCV, consistent with the cytoplasmic life cycle of HCV (*Lindenbach & Rice, 2001*).

The mature core protein consists of a N-terminal two-thirds hydrophilic domain of 120 aa or so and a C-terminal one-third hydrophobic domain of 50 aa or so (*McLauchlan, 2000*). Similar to the core proteins of HCV-related flaviviruses, the HCV core protein is essentially composed of $\alpha$-helices (50%). In the presence of mild detergents, it is a dimeric alpha-helical protein, but it forms heterogeneous soluble micelle-like aggregates of high molecular weight in the absence of detergents. In contrast, the C-terminal truncated form (117 residues) is soluble, monodispersed, and unfolded in the absence of detergent. The 117–169 hydrophobic domain contains the structural determinants for the HCV core protein binding to cellular membranes and is essential for the folding of the highly basic domain. Therefore, the hydrophobic C-terminal domain is responsible for core association with lipid droplets in mammalian cells and with endoplasmic reticulum membranes, and the N-terminal domain that includes numerous positively charged amino acids is mainly involved in RNA binding (*Boulant et al., 2005*).

*Kunkel & Watowich (2004)* showed that the C-terminal truncated HCV core protein of 124 amino acid residues (C124) forms a hydrophilic domain with three highly basic clusters, which presents disordered structure in solution, like other "intrinsically disordered" proteins ("IDPs"), characterized by a red shift in the tryptophan fluorescence spectra, which is consistent with exposed residues to polar environment, and negative ellipticity at 200 nm in circular dichroism (CD) spectra (*Uversky, 1999*; *Dunker et al., 2013*). Additionally, structure and dynamics of the HCV core protein (1–82 amino acids) by NMR showed that the N-terminal half is unstructured in aqueous solution and that it is a member of the IDP (Intrinsically Disordered Protein) family (*Duvignaud et al., 2009*). The expression of C-terminal truncated HCV core protein in *Escherichia coli* leads to the assembly of nucleocapsid-like particles (NLPs) (*Lorenzo et al., 2001*). In addition, C124 and other truncated forms of the core protein are able to interact with unspecific nucleic acids of different sizes, in micromolar range, and to assemble into NLPs *in vitro* (*Kunkel et al., 2001*; *Majeau et al., 2004*; *Acosta-Rivero et al., 2005*; *Fromentin et al., 2007*).

The focus of our research was to seek new structural and thermodynamic information to better understand the molecular aspects of the N-terminal region of core protein from C-terminal truncated HCV core protein (C124). Our data indicate that C124 has a low propensity for overall folding. In contrast, by electron microscopy, we show an unusual capacity of C124, at low concentration and in the absence of nucleic acids, to naturally multimerize into empty nucleocapsid-like particles (NLPs) when subjected to a pH close to its isoelectric point. Moreover, our data indicate that C124 is able to sequester a great number of unspecific nucleic acids of molecular size equivalent to the cellular microRNAs into NLPs in the nanomolar range. Our findings reveal features that can be related to the multiplicity of functions of HCV core protein, such as gene regulation, and explain why the *in vitro* formation of NLPs does not require high specificity, being mainly driven by neutralization of basic residues, which correspond to approximately 20% of the C-terminal truncated HCV core protein. Implications in virus-host interaction and HCV pathogenesis are discussed.

## MATERIALS & METHODS

### Chemicals

All reagents were of analytical grade. Distilled water was filtered and deionized through a Millipore water purification system. The probe bis-8-anilinonaphthalene-1-sulfonate (bis-ANS) was purchased from Invitrogen. All experiments were performed at 20 °C using the standard buffer: 10 mM phosphate (pH 7.0) with 100 mM NaCl.

### Nucleic acid samples

High pressure liquid chromatography-purified synthetic single-stranded RNA fragment 43–59 of SAF93 aptamer (SAF93[43–59]-5′-GGA UGC AAU CUC CAU CCC-3′) (*Rhie et al., 2003*) was obtained from Integrated DNA Technologies, Inc. (Coralville, IA, USA). Synthetic RNA samples were maintained lyophilized at −20 °C and used in RNase-free water. Double-stranded oligonucleotides were prepared by mixing equimolar amounts of the complementary single-stranded oligonucleotides, poly(GC) DNA (5′ ATAATTGCGCGCGCGCGCAGGAAA3′) (purchased from DNAgency, Malvern, PA) or consensus DNA (5′ TTTCCTAGACATGC-CTAATTA 3′) (purchased from Invitrogen, Carlsbad, CA, USA), in 50 mM Tris–HCl, pH 7.2, containing 250 mM NaCl. This mixture was incubated at 96 °C for 5 min, and the temperature was slowly reduced to 25 °C.

### Cloning and expression of the C-terminal truncated HCV core protein

We amplified the HCV core sequence by PCR from pCV-H77C, an infectious cDNA clone of type 1a (from J Bukh, NIH) (*Yanagi et al., 1997*), using a 5′ primer with the sequence 5′-GCGCCATATGAGCACGAATCCTAAACCT-3′ a 3′ primer of sequence 5′-GCGGATCCTCAGGCTGAAGCGGGCACAGTCAG-3′, and Vent DNA polymerase (New England Biolabs). The result was a DNA fragment encoding amino acids 1–124 of core protein with a NdeI site at its initiator AUG and a nonsense triplet at codon 125 followed immediately by a BamHI site. After digestion with NdeI and BamHI, the fragment was ligated to pET15b (from Novagen which harbors a 6-histidine tag at the C-terminal end to

ease the purification process on a nickel affinity column (Qiagen)) cleaved with the same enzymes, Biolabs. The C124 was propagated to midlog phase ($OD_{600} = 0.8$) in *Escherichia coli* strain BL21(DE3) at 25 °C. Expression of C124 fused to a histidine tail was induced with 1 mM IPTG. Three hours after induction the cells were centrifuged (5,500 g for 20 min) at 4 °C and frozen at −20 °C overnight.

## Purification of the C-terminal truncated HCV core protein

After thawing, the cells were ressuspended in lysis buffer (25 mM $NaH_2 PO_4$, 250 mM NaCl, 8 M urea, 2 mM EDTA and 2 mM DTT, pH 7.0), and were sonicated. The cell debris was pelleted by centrifugation (13,500 g for 20 min). The clarified lysate containing the core protein was applied to a cation-exchange column (SP Sepharose) equilibrated with denaturing cation buffer (25 mM Hepes, pH 7.0, 50 mM NaCl, 8 M urea). The core protein was eluted with a linear NaCl gradient (0.05–1.0 M). Fractions containing core protein were subsequently applied to a nickel-NTA agarose resin at pH 7.0, and to promote the elution we utilized the buffer (10 mM Tris, 100 mM NaCl) at pH 3.0. To the measurements in neutral conditions, the protein in high concentration was diluted in the same buffer at pH 7.0. The homogeneity of purified core protein was determined by 15% sodium dodecyl sulfate-polyacrylamide gel electrophoresis (SDS-PAGE) stained with Comassie Brilliant Blue. Protein concentration was determined by UV spectrophotometry at a wavelength of 280 nm in 6 M guanidine hydrochloride solution, using an extinction coefficient of 31,970 $M^{-1}$ determined using ExPASy program (*Appel, Bairoch & Hochstrasser, 1994*).

## Fluorescence spectroscopy

Fluorescence spectra were recorded in an ISSK2 spectrofluorometer (ISS Inc., Champaign, IL, USA). The protein was excited at 280 nm and emission was observed from 300 to 420 nm. For experiments in the presence of bis-ANS, the excitation wavelength was 360 nm and emission was collected from 400 nm to 600 nm.

## Light scattering

Light scattering (LS) measurements were performed in an ISSK2 spectrofluorometer (ISS Inc., Champaign, IL, USA). Scattered light was collected at an angle of 90° to the incident light. The samples were illuminated at 320 nm and the light was collected in the range of 315 to 325 nm. This wavelength was chosen because neither protein nor nucleic acid absorbs at 320 nm. In each pH, the LS values were obtained from the area under LS curve.

## Circular dichroism

Circular dichroism (CD) spectra of C124 were recorded on a Jasco J-715 (1505 model) spectropolarimeter (Jasco Corporation, Tokyo, Japan) with 0.02 cm circular path length cells at 20 °C. Wavelength range: 260–190 nm. The spectra were averaged from three scans that were recorded at 50 nm/min and are representative of three independent experiments. The samples were diluted for a 25 μM final concentration. All spectra had the spectrum of the buffer subtracted.

### Temporal evolution studies of *in vitro* assembly reactions

For temporal evolution studies, the optical density was monitored at 350 nm in a spectrophotometer at 20 °C. For analysis of NLPs formation promoted by unspecific nucleic acids, fixed amount of DNA (100 µL), to the final concentration of 5 µM, was added to different concentrations of C124 to a final volume of 600 µL. Optical density was recorded by Swift II software every 2 s for 5 min. The maximum value represents the maximum optical density obtained during this 5 min analysis.

### Transmission electron microscopy

After specific temporal studies the same samples were adsorbed to carbon coated Cu grids, 400 mesh, for 10 min. Grids were washed three times with filtered, distilled water and stained for 2 min with 2% uranyl acetate, and the NLPs were examined on a Morgani electron microscope (FEI Co., Hillsboro, OR, USA) operated at 100 kV. After scanning the negatives, images were processed with Adobe Photoshop (Adobe Systems Inc., Mountain View, CA).

### Isothermal titration calorimetry (ITC)

ITC measurements were performed using a VP-ITC calorimeter from MicroCal, Llc (Northampton, MA, USA). The titration of 20 µM HCV core protein with unspecific nucleic acids (RNA or DNA) involved 13 injections (1 × 1 µL and 12 × 20 µL) of 0.1 mM nucleic acid solution at 5 min intervals, with constant stirring at 317 rpm. The temperature was set at 37 °C. The C124 solutions were degassed under vacuum before the titrations, and the reference cell was filled with Milli-Q water. The heat of dilution of nucleic acids into the buffer was subtracted from the raw data obtained using C124. Three independent experiments were analyzed separately.

### Fluorescence correlation spectroscopy (FCS) measurements

FCS measurements were carried out in an ALBA fluorescence correlation spectrometer (ISS, Champaign, IL, USA) using a Nikon TE2000-U inverted microscope with a two-photon excitation regime. A Ti:Sa Tsunami laser, pumped by a Millenia Pro 15sJ (Spectra Physics, Mountain View, CA, USA), was focused on the sample with a water immersion 63_objective, 1.2 NA. A wavelength of 780 nm was used for excitation. The nucleic acids were FITC or Alexa-488-labeled. Fluorescence was collected through the same lens and separated from excitation by a dichroic mirror 700dcxru (Chroma, VT, USA). After passing through the dichroic mirror, the beam was split by a 50/50 beam splitter, and the resulting beams were focused to two APD detectors. Recorded fluctuation traces were processed in Vista ISS software for autocorrelation.

## RESULTS

### HCV C124 protein structure and the effect of alcohols

The C124 is the hydrophilic N-terminal domain of the HCV core protein and has been described as the region responsible by viral RNA interaction and assembly (Fig. 1). This region presents three highly basic clusters that are rich in arginine residues as showed in blue in the Fig. 1A. On the other hand, the C-terminal domain is rich in hydrophobic residues, as

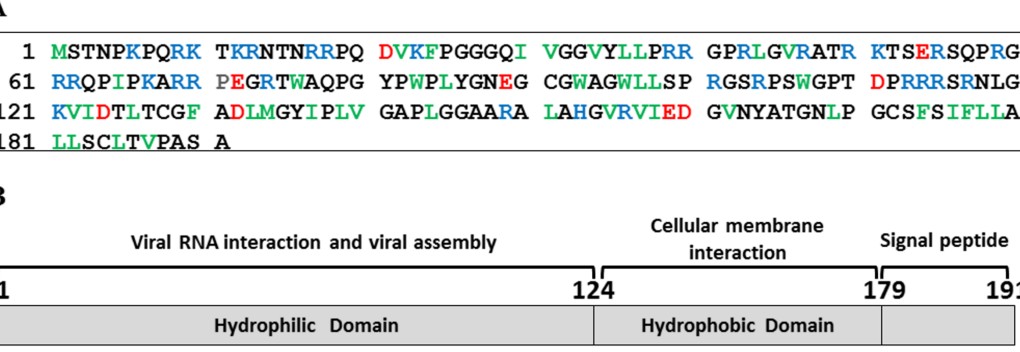

**A**

```
  1  MSTNPKPQRK  TKRNTNRRPQ  DVKFPGGGQI  VGGVYLLPRR  GPRLGVRATR  KTSERSQPRG
 61  RRQPIPKARR  PEGRTWAQPG  YPWPLYGNEG  CGWAGWLLSP  RGSRPSWGPT  DPRRRSRNLG
121  KVIDTLTCGF  ADLMGYIPLV  GAPLGGAARA  LAHGVRVIED  GVNYATGNLP  GCSFSIFLLA
181  LLSCLTVPAS  A
```

**B**

**Figure 1 HCV core protein.** (A) Primary sequence of HCV core protein. Hydrophobic residues (L, I, V, F, W and M) are shown in green, acidic residues in red and basic residues in blue. (B) Scheme of HCV core protein domains and their respective functions are highlighted.

shown in green, being important to the cellular membrane interaction (Fig. 1A), and the signal peptide direct hepatitis C virus polyprotein processing in the endoplasmic reticulum (Fig. 1B).

In order to analyze the C124 structure, we probed the intrinsic fluorescence of the six tryptophan residues, located at positions 76, 83, 93, 96, 107 and 113 of the protein. Our data showed that these residues were extremely solvent-exposed (Fig. 1A) as evidenced by a center of spectral mass of 352 nm, reflecting characteristics of an disordered protein. The far-UV CD spectrum of C124 exhibited a negative signal at 200 nm, in agreement with a non-structured protein (Fig. 1B).

Structural studies in the presence of different solvents can provide information about the stability, folding pathway and intermolecular interactions of a protein. Alcohols are known to modulate the interactions of the polypeptide chain, and trifluorethanol (TFE) has been used to induce the formation of helical structure in protein fragments and peptides (*Dyson & Wright, 1993*; *Hamada et al., 2000*) and in many other instances is known to transform proteins into molten globule-like intermediates (*Konno, Iwashita & Nagayama, 2000*) and sometimes to stabilize intermediate structures (*Luo & Baldwin, 1998*). To investigate the effects of alcohols on the C124 structure, the samples were incubated in the presence of 10–50% of either butanol or its fluorinated derivative, TFE. The most significant effects were observed at 30% of both alcohols and are shown in Fig. 2. The intrinsic fluorescence spectrum (Fig. 2A—dotted line, TFE, dashed line, butanol) underwent a slight blue shift in the presence of the alcohols, indicating little change in the tertiary structure. The spectral center of mass of the C124 fluorescence spectrum changed from 343 nm to 341 nm in the presence of 30% TFE and to 337 nm in the presence of 30% butanol. Although small changes in tryptophan environment were observed, the results showed that TFE was able to promote significant alterations in the secondary structure of C124, as indicated by the effects on the circular dichroism spectrum (Fig. 2B, dotted line).

## Analysis of bis-ANS binding to C124

As part of the structural characterization of C124 we used the fluorophore bis-ANS (bis-8-anilinonaphthalene-1-sulfonate), which binds noncovalently to nonpolar segments

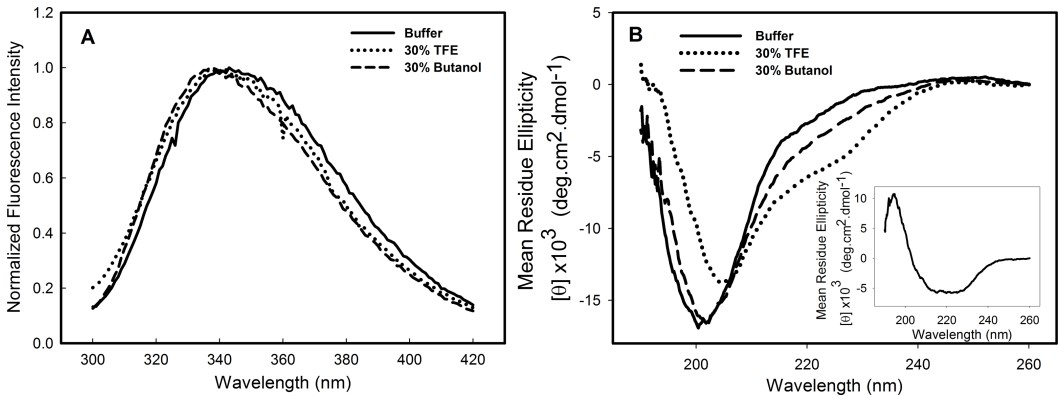

**Figure 2  Structural changes of HCV C124 induced by TFE and butanol.** Changes on the tertiary structure (A) and secondary structure (B) of C124 (25 μM) in the presence of 30% butanol or 30% TFE. Inset shows the resulting CD spectrum obtained after subtracting the C124 CD spectrum in the presence of TFE from that in the absence of TFE.

in proteins, especially in proximity to positive charges (*Rosen & Weber, 1969*; *Silva et al., 1992*). Since its binding is followed by a large increase in its fluorescence quantum yield, this probe has been frequently used to investigate the presence of hydrophobic sites, conformational changes, and to detect the presence of molten globule states of proteins (*Silva et al., 1992*). Additionally, the effects of this fluorescent probe on protein conformation and stability have been studied (*Shi, Palleros & Fink, 1994*; *Foguel et al., 1996*; *Lima et al., 2006*). Here, we show that bis-ANS was able to bind to C124, as verified by the increase on bis-ANS fluorescence emission intensity (Fig. 3A). Nevertheless, binding of the probe did not significantly change either the CD (Fig. 3B) or the intrinsic fluorescence spectra (Inset, Fig. 3B), indicating no significant alterations in the secondary structure or in the tryptophan environment of C124 (Fig. 3B). We were not able to observe any increase in turbidity or in light scattering in the presence of bis-ANS or alcohols (data not shown). IDPs CD spectra may be derived of at least three different disordered equilibrium conformations, molten globule (MG), premolten globule (PMG), and random coil (RC) (*Uversky, 2002*; *Habchi et al., 2014*). The PMG-like and RD-like forms can be subdivided from the correlation between $\theta_{222}$ and $\theta_{200}$ values, as described by *Uversky (2002)* (Fig. 3C). This analysis shows that C124 is a random coil in solution (Fig. 3C) as confirmed by the analysis of the ratio between the $\theta_{222}$ and $\theta_{200}$ (Fig. 3D), as described by *Blocquel et al. (2012)*. However, the ratio $\theta_{222}/\theta_{200}$ of C124 CD spectrum in the presence of bis-ANS changed to 0.217 suggesting that bis-ANS binding promotes a transition of a RC-like to a PMG-like form (Fig. 3D). The $\theta_{222}/\theta_{200}$ analysis has the advantage since it undergoes smaller interference of errors in estimations of protein concentrations (*Blocquel et al., 2012*).

## Changes in pH induce the formation of empty nucleocapsid-like particles (NLPs)

Changes in pH can induce partial folding of intrinsically disordered proteins due to the minimization of their large net charge at neutral pH, decrease of charge/charge intramolecular repulsion and the hydrophobic-driven collapse (*Ahmad et al., 2010*). Our results
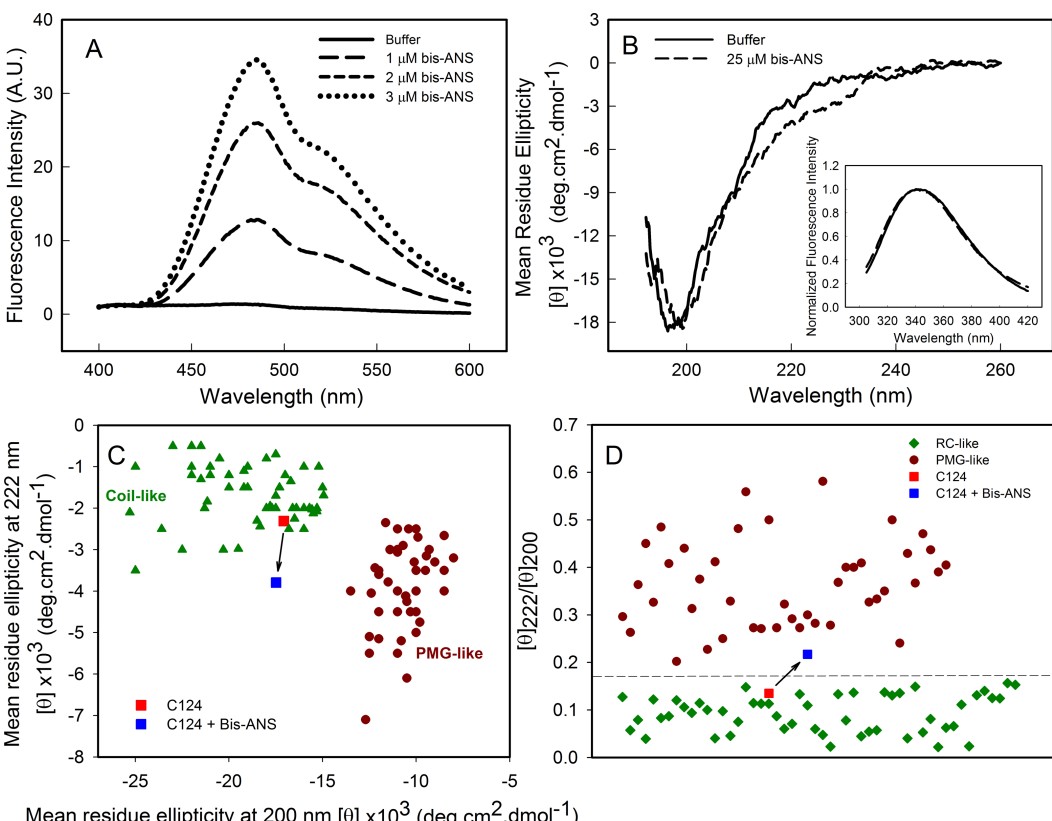

**Figure 3** **Analysis of binding of bis-ANS to C124.** (A) Analysis of bis-ANS binding to C124 (1 μM) by the increase of fluorescence intensity of bis-ANS spectra. (B) Changes on the secondary structure observed by Far-UV CD of C124 (25 μM) in the presence of the probe (25 μM). Inset: intrinsic fluorescence measurements of C124 in the absence and in the presence of bis-ANS at room temperature. (C) Double wavelength plot, $[\theta]_{222}$ versus $[\theta]_{200}$, modified from *Uversky (2002)*, of a set of well-characterized unfolded, RC-like (dark green diamonds) or PMG-like proteins (dark red circles), and of the C124 in the absence or presence of Bis-ANS (25 μM) that the positions are highlighted (red and blue squares, respectively). (D) Plot of the ratio between the ellipticity at 222 nm and the ellipticity at 200 nm ($[\theta]_{222}/[\theta]_{200}$) of the same set of well-characterized RC-like or PMG-like proteins shown in (C). The position in the plot of C124 in the absence or presence of bis-ANS is highlighted (red and blue squares, respectively). The arrows in (C) and (D) are indicating the changes in the C124 position promoted by bis-ANS binding.

showed that the decrease in pH did not promote structural changes on the C124 structure. In contrast, the increase in pH to near the pI of C124 (pH ≈ 12) caused a slight blue shift in the spectra indicating small changes in the tertiary structure (Fig. 4A). Additionally, an increase in intensity was observed on the left side of the emission spectrum at pH 11, indicative of excitation light scattering and bleed-through. Therefore, we also observed an increase in light scattering at 320 nm when the pH was raised to 11 and up (Fig. 4B), suggesting protein aggregation or NLP assembly, in the absence of nucleic acids. The temporal evolution of the NLPs assembly by spectrophotometric measurements showed a stabilization between 100 and 300 s at conditions utilized since the OD remained constant (Fig. 4C). Kinetics analysis was not possible because this process appears to occur in a subsecond time scale. In addition, our data also showed that the increase on protein

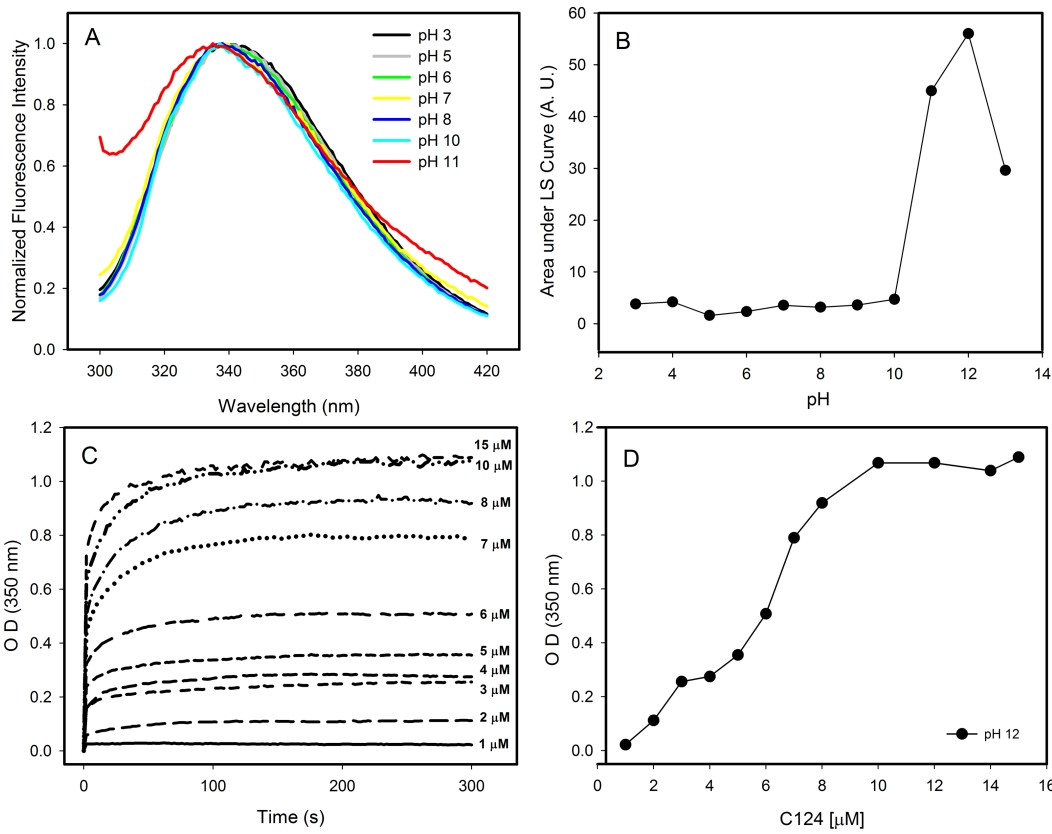

**Figure 4** **Effects of pH on the structure of C124.** (A) Changes in the intrinsic fluorescence induced by high pH values. (B) Light scattering of C124 at different pH values. (C) Temporal evolution analysis of *in vitro* assembly of NLPs at different concentrations of C124 triggered by the pH 12. (D) Plot of the maximum O.D. values at 350 nm as derived from the curves in (C) at different C124 concentrations.

concentration correlated with an increase of optical density, as expected to protein self-assembly process (Fig. 4D).

Images obtained by transmission electron microscopy confirmed the formation of NLPs at pH 12 (Fig. 5). The micrographs showed the formation of NLPs with heterogeneous particle size at pH 12, as already observed for NLP assembly induced by some nucleic acids (*Kunkel et al., 2001*).

## The process of NLPs assembly is triggered by short unspecific nucleic acids

The formation of NLPs triggered by addition of nucleic acids has been extensively characterized (*Fromentin et al., 2007*; *Kunkel et al., 2001*). Here, we used other methodologies to better understand the interaction of nucleic acids and C124 and the NLPs assembly. Accordingly, we selected unspecific nucleic acids with equivalent size to cellular microRNAs with the aim of better understanding the possible direct interaction of the core protein with cellular RNA. Previous work described a method to follow the kinetics of *in vitro* assembly by measuring the solution turbidity using a spectrophotometer (*Fromentin et al., 2007*). We performed similar experiments in order to investigate if three short

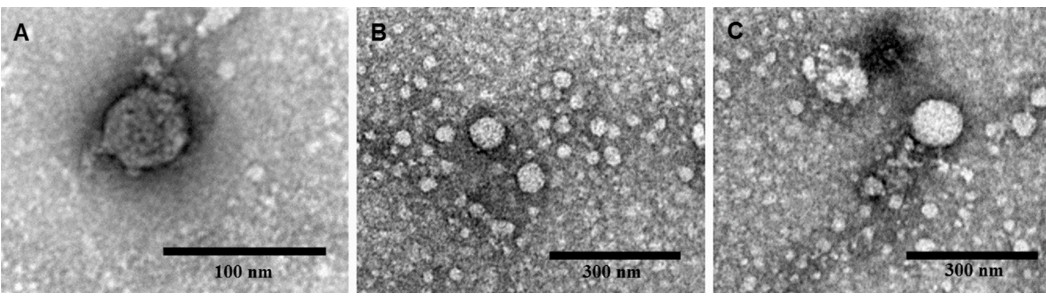

**Figure 5** **Electron micrographs of negatively stained nucleocapsid-like particles (NLPs) produced from truncated HCV core protein at pH 12.** The protein concentration was 20 μM. Bars: 100 nm (A) and 300 nm (B and C).

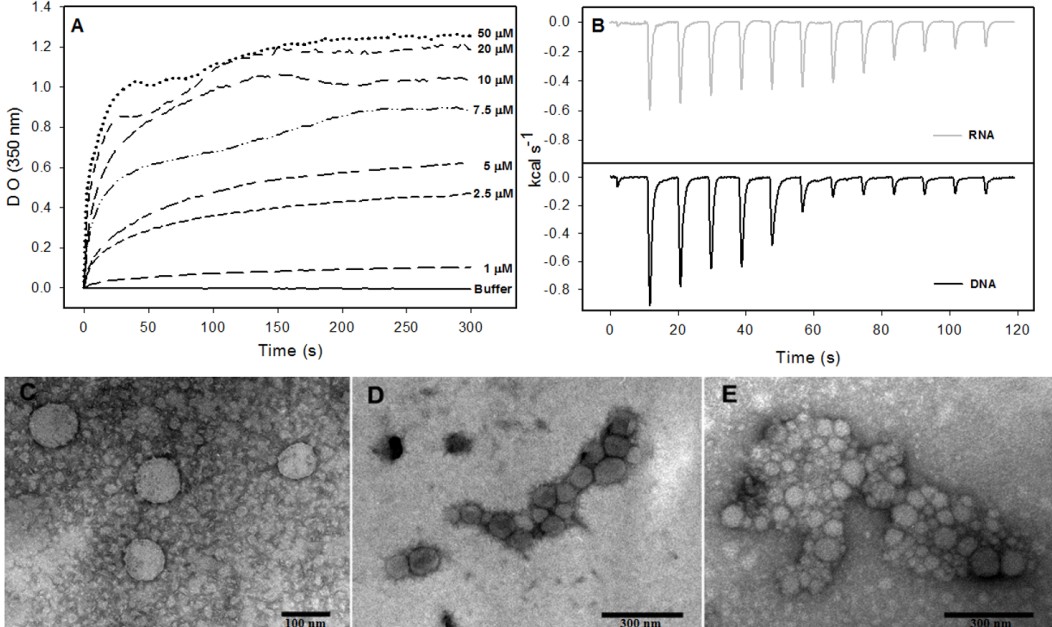

**Figure 6** **Interaction of HCV core protein with unspecific nucleic acids and NLPs formation.** (A) Temporal evolution studies of *in vitro* assembly of NLPs at different concentrations of C124 triggered by the addition of 5 μM DNA poly(GC). (B) Heat flux profile associated with injections of 5 μM nucleic acid (poly(GC) DNA or RNA (SAF93[43–59])) in the calorimetric cell containing C124 (20 μM) at 37 °C. (C–E) Electron micrograph of negatively stained nucleocapsid-like particles (NLPs) produced from C124 at 2.5 μM (C), 10 μM (D) and 50 μM (E). To each protein concentration was added 5 μM poly(GC) DNA. Bars: (C)—100 nm and (D, E)—300 nm.

unspecific nucleic acids, DNA poly(GC) and p53 consensus (both with 21 bp), and a short structured RNA (18 bp), were able to trigger NLP assembly. Here we show that both DNAs and RNA were substrates for the formation of NLPs. This process was dependent on the concentrations of both DNA or RNA and of protein as verified by spectrophotometry (Fig. 6A and Fig. S1) and confirmed by electron microscopy (Figs. 6A–6E). We observed NLP aggregation when the assay was carried out with high protein concentration (Figs. 6D–6E).

### The process of NLPs assembly triggered by nucleic acids is enthalpy driven

Isothermal titration calorimetry (ITC) is often used to measure the heat absorbed or released during a reaction and, thus, can provide a universal means to follow biological processes (*Ladbury, 2004*). We used ITC to investigate the energetics of NLP assembly induced by the interaction of C124 with nonspecific nucleic acids. As shown in the calorimetric traces in Fig. 6B, each injection of nucleic acids (DNA or RNA) into the protein solution resulted in an exothermic reaction, showing that nucleic acid-triggered capsid assembly in solution is enthalpically driven at 37 °C (Fig. 6B). These data indicate an important role of non-hydrophobic interactions, such as the electrostatic interactions between basic residues of the protein and phosphate groups on the nucleic acids, and suggest that charge neutralization plays an important role in the particle assembly process. This is also consistent with our results showing capsid assembly at pH = 12. These may be the main interactions driving the assembly process *in vitro*.

### Fluorescence correlation spectroscopy analysis indicates capsid assembly with unspecific nucleic acid in nanomolar range

Although the capsid assembly promoted by unspecific nucleic acids has been studied (*Fromentin et al., 2007*; *Kunkel et al., 2001*), the concentrations of both protein and nucleic acid used in most of that work were in the micromolar range. In the experiments described here, we applied fluorescence correlation spectroscopy (FCS) to gain new information on the interaction between nucleic acids and the C-terminal truncated HCV core protein in the nanomolar range, by using DNA or RNA labeled with Alexa-488.

Observing the raw data from FCS measurements we verified that the fluorescence fluctuation from the free RNA is very homogeneous (Fig. 7A). The results were similar to RNA (Fig. 7A) and DNA (Fig. S2). On the other hand, in the presence of higher concentrations of HCV core protein we observed rare fluorescent species with very high fluorescence intensity (Fig. 7B). In addition, the signal of the free nucleic acids basically disappeared, presumably due to RNA oligomerization promoted by capsid assembly (Fig. 7B). These data suggest that the assembly is a highly cooperative process since stable intermediates were not observed. For comparison of the diffusion times in DNA or RNA in the absence or in the presence of C124, the normalized autocorrelation curves are shown in Fig. 7C. The protein promoted a great increase in the diffusion time of both RNA and DNA, consistent with the formation of NLPs.

## DISCUSSION

Here, we have demonstrated that the C-terminal truncated HCV core protein has a high intrinsic and self-sufficient propensity to assembly into NLPs, but with low propensity for intramolecular folding. The loss of positive charge (Figs. 4 and 5) or nonspecific neutralization (Figs. 6 and 7) of basic residues of the HCV core protein appears to be the major factor that drives the *in vitro* formation of NLP. Our findings suggest that the excess of positive charges of C124 can represent an important energetic barrier to the inherent process of protein multimerization into empty NLPs (Fig. 8). Additionally, the protein's ability

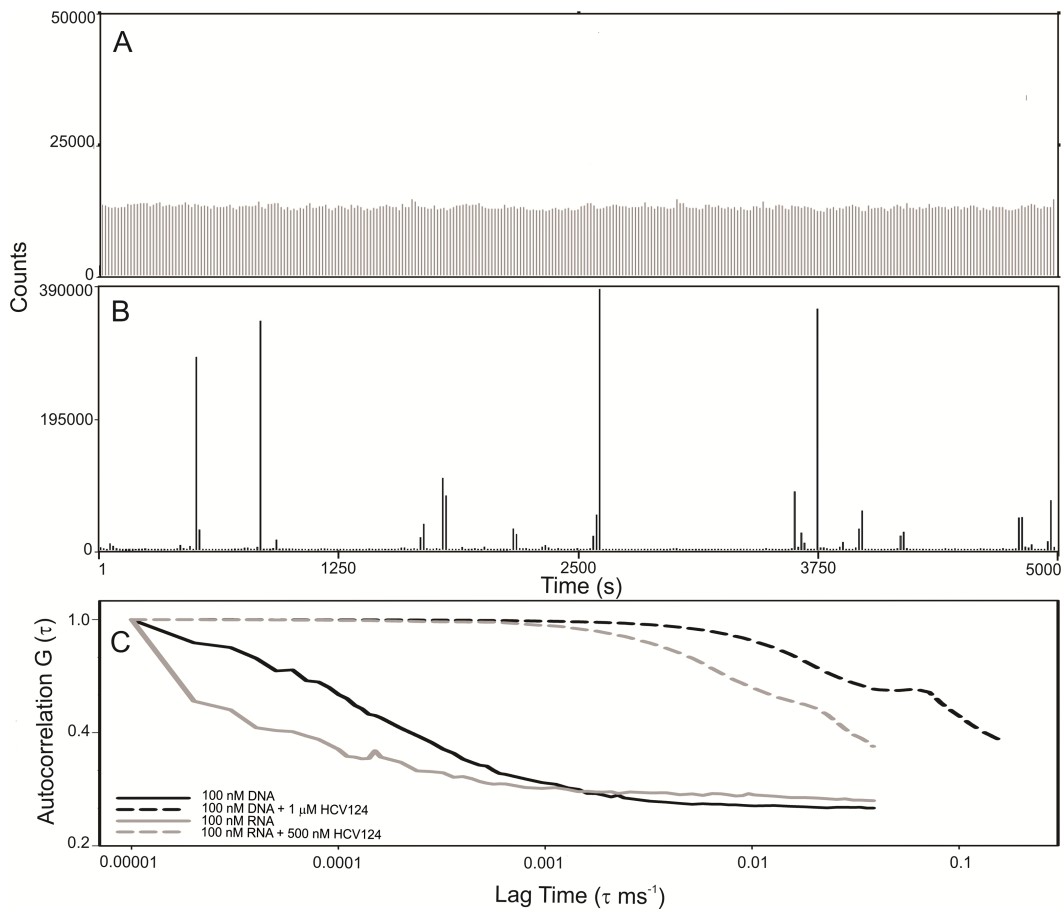

**Figure 7  C-terminal truncated HCV core protein and unspecific nucleic acids (DNA or RNA) interaction analyzed by fluorescence correlation spectroscopy.** Fluctuation of the fluorescence intensity of RNA labeled with Alexa-488 in the absence (A) or in the presence (B) of C-terminal truncated HCV core protein, and the normalized autocorrelation curves of free DNA or RNA at 100 nM and in the presence of 1 µM or 500 nM of C124 (C). The buffer used was 10 mM phosphate (pH 7.4) with 100 mM NaCl.

to nonspecifically package nucleic acids could have important pathological implications. The positive charges on the protein subunits are expected to act against the assembly of capsids due to electrostatic repulsion (*Siber & Podgornik, 2007*). Previous work showed that a C-terminal truncated Hepatitis C Virus core protein variant, different from the protein used in this work, assembles into virus-like particles *in vitro* in the absence of structured nucleic acids, however only at very high concentrations (*Acosta-Rivero et al., 2005*). These data are consistent with our suggestion that C124 has a self-assembly tendency, with the positive charges of basic residues representing an energy barrier to NLP formation. The simple loss of these charges, for example by deprotonation at pH close to 12 (Figs. 4 and 5), favors the formation of contacts among C124 monomers necessary to the assembly process (*Rodríguez-Casado, Molina & Carmona, 2006*; *Rodríguez-Casado, Molina & Carmona, 2007*). This characteristic also explains the lack of specificity of nucleic acid packaging *in vitro* (*Fromentin et al., 2007*; *Kunkel et al., 2001*).

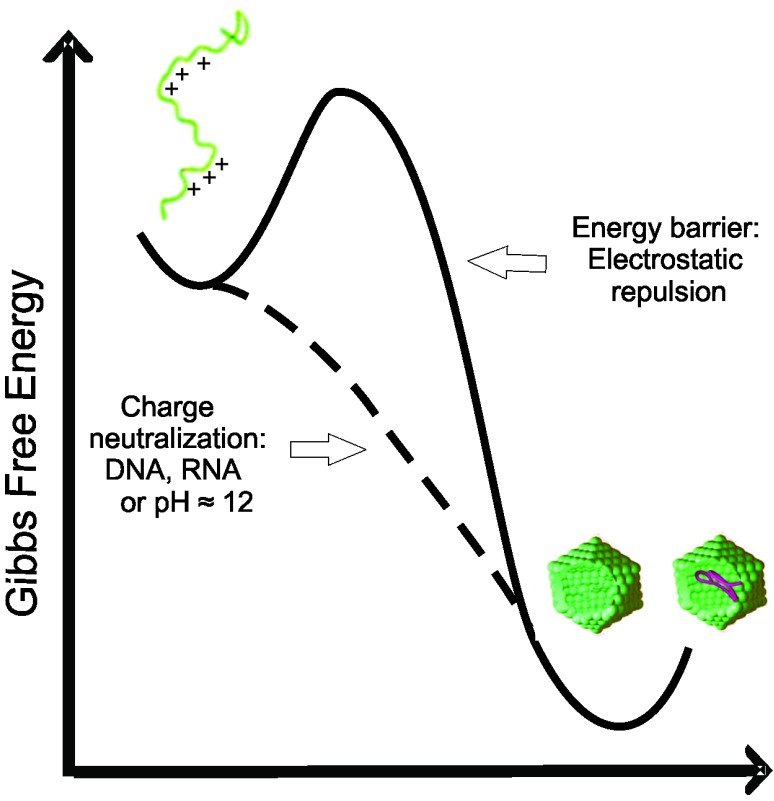

**Figure 8** Free energy diagram representing the energy barrier between the disordered state of C124 to oligomeric state (empty capsid or nucleic acid loaded capsid).

The multiplicity of HCV core protein function may be related, at least in part, to the fact that the N-terminal half of this protein (C82) is an intrinsically disordered protein (IDP) domain (*Duvignaud et al., 2009*) The lack of rigid globular structure under physiological conditions might represent a considerable functional advantage allowing this protein to interact efficiently with different targets in the cell (*Dyson & Wright, 2005*; *Tompa, 2005*; *Habchi et al., 2014*). Additionally, IDP domains can adopt secondary structure when interacting with other molecules, such as nucleic acids or proteins. It is possible to mimic these conditions by modifying the environment of the protein. Several studies have shown the influence of cosolvents on the folding kinetics of proteins and peptides (*Hamada et al., 2000*; *Konno, Iwashita & Nagayama, 2000*; *Luo & Baldwin, 1998*). As monitored by far-UV CD, we showed that only TFE was able to promote significant change in the secondary structure of C124 (Fig. 2B). Consistent with our observation, previously published NMR data showed that the N-terminal half (1–82 residues) of core protein can adopt an alpha-helical structure in the presence of TFE (*Duvignaud, Leclerc & Gagné, 2010*), and this conformation has been proposed to be relevant to one of the functional roles of the HCV core protein. Other conditions, such as in the presence of SDS monomers and micelles, or different concentrations of NaCl (Fig. S3), do not promote significant changes in C124 structure, suggesting that C124 does not usually fold into a well-defined secondary structure.

Although bis-ANS was able to induce only subtle changes of protein secondary structure, the binding verified by the increase of fluorescence intensity indicated the presence of structured hydrophobic regions in the C-terminal truncated HCV core protein (Fig. 3A). Bis-ANS binds noncovalently to nonpolar segments in proteins, especially in proximity to positive charges (*Rosen & Weber, 1969*; *Silva et al., 1992*). Proteins that consist of disordered regions interspersed with short structured regions are typically intrinsically disordered proteins. There are at least three different disordered equilibrium conformations, molten globule, premolten globule, and randon coil in IDPs (*Uversky, 2002*; *Habchi et al., 2014*). Analysis of the correlation between $\theta_{200}$ and $\theta_{222}$ values (*Uversky, 2002*) and ratio between $\theta_{222}$ and $\theta_{200}$ (*Blocquel et al., 2012*) indicates that C124 is RC-like in solution (Figs. 3C and 3D). Additionally, the change of $\theta_{222}/\theta_{200} = 0.135$ to $\theta_{222}/\theta_{200} = 0.217$ indicates that bis-ANS binding promotes the transition of a RC-like to a PMG-like form possibly by folding intermediate stabilization. Additionally, consistent with this finding, previous studies described that hydrophobic fluorescence probe ANS also is able to interact with premolten globule state in proteins (*Uversky & Ptitsyn, 1996*; *Uversky, 2002*). These structured regions are known to be important in specific ligand binding in some cases (*Fuxreiter et al., 2004*; *Mohan et al., 2006*; *Habchi et al., 2014*). The stabilization of PMG-like form in the C124 structure, as indicated by bis-ANS binding analysis, may also account for the ability of binding to different cellular targets.

Energetic aspects of the capsid assembly are important to the understanding of the biological process. The assembly of Hepatitis B Virus capsids is driven by weak protein-protein interactions and is characterized by positive enthalpy and entropy (*Ceres & Zlotnick, 2002*). This assembly process is entropy-driven, and is characterized largely by hydrophobic contacts. On the other hand, the binding of intrinsically disordered proteins, which are usually highly hydrophilic, involves an entropic cost associated with the disorder-to-order transition (*Dyson & Wright, 2005*; *Habchi et al., 2014*). Here, we verified that a negative enthalpic contribution is the key thermodynamic driving force for NLP assembly by C124 (Fig. 6B). It is likely that the negative enthalpy offsets the entropic cost, giving a good example of enthalpy-entropy compensation associated to assembly process. The heat released in the assembly process is probably mainly due to the neutralization of basic residues as they interact electrostatically with nucleic acids.

Neither the assembly pathway nor the structure of the HCV nucleocapsid and virion *in vivo* have been completely elucidated. As for other flaviviruses, HCV core protein interacts with nucleic acids *in vitro* to form nucleocapsid particles. However, the degree of specificity for nucleic acids necessary to assembly into NLPs is not well characterized. Some works showed a low specificity of nucleic acid packaging by core protein *in vitro* (*Lorenzo et al., 2001*; *Fromentin et al., 2007*; *Kunkel et al., 2001*). *Lorenzo et al. (2001)*, reported that VLPs are generated when the first 120 aa of HCV core protein are expressed in *E. coli* indicating also low specificity necessary to NLPs assembly. We observed that C124 naturally multimerizes into empty nucleocapsid-like particles (NLPs) when subjected to a pH close to its isoelectric point (pH = 12) (Figs. 4 and 5) support that the charge neutralization is the major factor leading to the protein assembly. Similar to what we observed for HCV core protein, the capsid dimer of Dengue virus forms particles independently of the sequence

and length of the DNA used, suggesting that the neutralization of positive charges might be the main event driving the assembly process (*López et al., 2009*). The fact that all these studies were performed in micromolar range and the binding affinities are not well characterized must be given due importance.

On the other hand, both gel mobility shift assay (*Tanaka et al., 2000*) and surface plasmon resonance (*Fan et al., 1999*) suggested specific binding of HCV core protein to synthetic oligonucleotides corresponding to the 59 or 58 untranslated region of the viral genome. These studies with the purified truncated HCV protein add important information, but did not simulate the real cellular conditions that are necessary to HCV replication. The assembly of native HCV particles in cells is extremely complex and research in the past few years has shown that formation of HCV-virions is closely connected to lipid droplets, which could serve as an assembly platform. It has also been shown that viral nonstructural proteins, such as NS2, NS3 and NS5, play key roles in HCV morphogenesis (*Boson et al., 2011*; *Suzuki, 2011*; *Bartenschlager et al., 2011*; *Counihan, Rawlinson & Lindenbach, 2011*). Moreover, HCV particle production appears to be strictly linked to very-low-density lipoproteins. So, the basis for core protein's specificity for the hepatitis C viral genome is poorly understood.

The C-terminal portion is crucial to directing the core protein to lipid droplets and for the production of infectious particles. However, the different aspects of the core protein in HCV replication and pathogenesis (*Kao, Yi & Huang, 2016*) can be better understood from the knowledge of the properties of the N-terminal portion. FCS data indicate that short unspecific nucleic acids, in the nanomolar range, undergo oligomerization in the presence of C124 (Fig. 7). This result suggests that, energetically, the formation of NLPs is highly favored and that in each NLP there is a great number of short unspecific nucleic acids. This assembly process could be promoted mainly by electrostatic interactions with nucleic acids (Fig. 6) or by simple deprotonation of basic residues triggered by higher pH (Figs. 4 and 5). In accordance with these findings, we are currently investigating the effects of heparin binding, a glycosaminoglycan with high negative charge density, and we also observed that it is able to promote an increase in turbidity indicating NLPs formation. Therefore, we suggest that the assembly process into NLPs by the C-terminal truncated HCV core protein does not require high specificity for viral RNA. Additionally, although the full-length protein and/or other co-factors, i.e., cellular chaperones, might be involved in determining the specificity of binding to specific nuclei acids and they are not considered in the present study, we speculate that core protein can interact with cellular polyanions, such as cellular mRNAs or microRNAs, triggering their confinement into NLPs or infectious particles. The shape and size of cellular RNA could be an important factor that determines the affinity, and consequently, preferential package leading to selective control of cellular microRNA and mRNA concentration. Some groups have previously shown microRNA deregulation in hepatitis C virus-infected human hepatoma cells (*Liu et al., 2010*; *Steuerwald et al., 2010*; *Conrad & Niepmann, 2014*; *Kao, Yi & Huang, 2016*), but the mechanisms are still not understood. Recent reports have showed that the interaction of miR-122 with the HCV genome is crucial for the accumulation of viral RNA in cultured liver cells (*Jopling et al., 2005*; *Conrad & Niepmann, 2014*; *Douam, Ding & Ploss, 2016*). Additionally, the sequestration of miR-122 by the HCV RNA induces a global de-repression

of miR-122 targets over the human transcriptome favoring the unbalance liver homeostasis and the development of liver cancers (*Luna et al., 2015*). We suggest that core protein could contribute to the deregulation of microRNA homeostasis from encapsulation of unspecific microRNAs or mRNA into infectious particles.

Recent studies have demonstrated that, besides the viral genome, host RNAs, such as mRNA and non-coding RNAs, are encapsidated by authentic flock house virus virions and virus-like particles. This important finding showed that, although there is high specificity for the viral RNA, a variable genetic content may also be packed by nonenveloped RNA viruses (*Routh, Domitrovic & Johnson, 2012a*; *Routh, Domitrovic & Johnson, 2012b*). One site containing three conserved phenylalanine residues at the C-terminal of the coat protein alpha of Flock House Virus has been identified as essential for the specific encapsidation of viral RNA. Deletion of all three residues almost totally abolishes viral RNA encapsidation, resulting in particles that primarily package cellular RNA (*Schneemann & Marshall, 1998*). These findings are comparable to the unspecific RNA packaging by the C-terminal truncated HCV core protein described here. Therefore, we speculate that similar events could occur during the production of HCV virions.

## CONCLUSIONS

The present study shows that the N-terminal of HCV core protein has low propensity to overall folding. However, in the absence of nucleic acids, it multimerizes itself into empty NLPs when submitted to pH values close to the isoelectric point (pH $\approx$ 12). This finding suggests that electrostatic repulsion among the positive charges of the basic residues of the C124, which represents about 20% of their residues, is the only energy barrier that avoids protein multimerization, and that unspecific polyanions could promote the formation of NLPs. Our observations suggest that C124 can physically interact with different polyanions, triggering their confinement into NLPs. It also explains why the *in vitro* NLP formation does not require high specificity. Based on these new findings, we speculate that this process could also happen in a host cell infected by HCV, promoting the confinement of microRNAs and mRNAs into infectious particles. We believe that the new data here obtained provide advances in the understanding of the molecular basis of some secondary effects of the core protein on the HCV pathogenesis.

## ACKNOWLEDGEMENTS

We gratefully acknowledge Emerson R. Gonçalves and Ana Carolina Q. Vaz for competent technical assistance.

### Funding

This work was supported by grants from Conselho Nacional de Desenvolvimento Científico e Tecnológico (CNPq), Coordenação de Aperfeiçoamento de Pessoal de Nível Superior (CAPES), Fundação Carlos Chagas Filho de Amparo à Pesquisa do Estado do Rio de

Janeiro (FAPERJ), Fundação Universitária José Bonifácio (FUJB), Instituto Milênio de Biologia Estrutural em Biomedicina e Biotecnologia (IMBEBB), Instituto Nacional de Ciência e Tecnologia de Biologia Estrutural e Bioimagem (INBEB), and Programa de Apoio a Núcleos de Excelência (PRONEX) to JLS, AMOG, and ACO. The funders had no role in study design, data collection and analysis, decision to publish, or preparation of the manuscript.

### Grant Disclosures

The following grant information was disclosed by the authors:
Conselho Nacional de Desenvolvimento Científico e Tecnológico (CNPq).
Coordenação de Aperfeiçoamento de Pessoal de Nível Superior (CAPES).
Fundação Carlos Chagas Filho de Amparo à Pesquisa do Estado do Rio de Janeiro (FAPERJ).
Fundação Universitária José Bonifácio (FUJB).
Instituto Milênio de Biologia Estrutural em Biomedicina e Biotecnologia (IMBEBB).
Instituto Nacional de Ciência e Tecnologia de Biologia Estrutural e Bioimagem (INBEB).
Programa de Apoio a Núcleos de Excelência (PRONEX).

### Competing Interests

Jerson Lima Silva is an Academic Editor for PeerJ.

### Author Contributions

- Theo Luiz Ferraz de Souza and Sheila Maria Barbosa de Lima conceived and designed the experiments, performed the experiments, analyzed the data, wrote the paper, prepared figures and/or tables, reviewed drafts of the paper.
- Vanessa L. de Azevedo Braga conceived and designed the experiments, performed the experiments, analyzed the data, prepared figures and/or tables, reviewed drafts of the paper.
- David S. Peabody conceived and designed the experiments, performed the experiments, analyzed the data, contributed reagents/materials/analysis tools, reviewed drafts of the paper.
- Davis Fernandes Ferreira performed the experiments, analyzed the data, reviewed drafts of the paper.
- M. Lucia Bianconi analyzed the data, reviewed drafts of the paper.
- Andre Marco de Oliveira Gomes conceived and designed the experiments, performed the experiments, analyzed the data, contributed reagents/materials/analysis tools, wrote the paper, reviewed drafts of the paper.
- Jerson Lima Silva analyzed the data, contributed reagents/materials/analysis tools, wrote the paper, reviewed drafts of the paper.
- Andréa Cheble de Oliveira conceived and designed the experiments, performed the experiments, analyzed the data, contributed reagents/materials/analysis tools, wrote the paper, prepared figures and/or tables, reviewed drafts of the paper.

### Data Availability

The raw data is included in the figures in the manuscript.

## Supplemental Information

Supplemental information for this article can be found online at http://dx.doi.org/10.7717/peerj.2670#supplemental-information.

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
