# Peer review of "Charge neutralization as the major factor for the assembly of nucleocapsid-like particles from C-terminal truncated hepatitis C virus core protein"

_PeerJ, doi:10.7717/peerj.2670_

## Round 0.1 · original submission · Major Revisions

Please address all the critical points raised by both reviewers. Please pay close attention to the suggestions provided by the reviewer #1.

Reviewer 1 ·

Basic reporting

Professional english should be improved
Text is clear unless stated

Experimental design

Methods described with sufficient details

Validity of the findings

Data is sound and conclusions are well stated, unless specified

Additional comments

The authors provide evidence that charge neutralization is key to promote the self-assembly of the HCV core protein into nucleocapsid-like particles (NLPs), which is otherwise intrinsically disordered. This finding is supported by a combination of biophysical studies, such as CD and fluorescence measurements, that showed a spontaneous self-assembly of the protein either at pH values close to its isoelectric point or in the presence of unspecific, non related nucleic acid sequences, thus suggesting that in both cases the electrostatic repulsion is reduced, which prompts the protein self-assembly.

I have no major objection with the paper content and the experimental procedures. The paper is well written, although needs improvements especially in the discussion part. The introduction is concise and clear and the experiments can be reproduced based on the material and methods.
I am therefore in favor of having this paper published in Peer J, in case the below points are properly addressed and which I believe can improve the paper.
- Many terms have been used throughout the manuscript to describe intrinsically disordered proteins. For less ambiguity and more consistency with the field, I think a unique terminology should be used, which was adopted by the field and is: intrinsically disordered proteins (IDPs) (see What’s in a name? Why these proteins are intrinsically disordered, Intrinsically Disordered Proteins, 2013).
- Many relevant references to the IDPs field are actually missing, such as Habchi et al. Chem Rev. 2014 and references therein cited
- I think it would be better to place the paragraph in the introduction starting at line 145 after line 110.
- It would be very helpful to have a figure 1 that shows the actual modular organization of the HCV core protein, with the figure indicating the residues using a color code for polarity/hydrophobicity, pre-identified binding sites, the truncated forms used in the present study, etc...
- In page 3, line 135, please clarify that “red shift in the tryptophan fluorescence…” is consistent with exposed residues to polar environment to make it clearer.
- It would be helpful to clarify why the pH for the Nickel affinity purification is 3, whereas all experiments were performed at pH7?
- Could you explain why when expressing the protein in E. coli, it does not form NLPs since in E. coli have nucleic acids? How is it possible to purify the monomeric species?
- I think adding the values of the maximum fluorescence at ~350 nm in Fig. 1A +/- the % of the relative change, i.e. blue shift, as a function of the addition of alcohol would be helpful as the difference is very small to be seen directly from the curves.
- In Fig. 3A, it is very difficult to match the curves with the conditions, please change to symbols.
- Please specify what is the “light scattering ratio” in Fig. 3B.
- Please clarify why light scattering experiments were performed at 320 nm (Fig. 3B), whereas the turbidity measurements (Fig. 3C) at 350 nm.
- Legend Fig. 3, line 360: please correct after “at 350 nm…” as derived from the curves in C at different C124 concentrations.
- Please specify in the text or/and in M&M that the kinetics in Fig. 3 are performed without DNA as according to the M&M all kinetics are performed with 5 μM DNA and that makes it confusing.
- I think the kinetics experiments merit to be discussed in more details. For instance, I can’t see any difference in the time that is required for the monomeric peptides to form the NLPs irrespective of the protein concentration. The difference is however in the intensity, which is directly related to protein concentrations and it is actually expected to have more signal resulting from the self-assembly of more proteins.
Does this mean that the self-assembly of the protein is independent from the protein concentrations? This is surprising as unlike protein folding that is a protein concentration-independent process, protein self-assembly is directly dependent on the concentration. Discussing this point would be helpful especially that this technique might also not be well-suited to study these kinetics as they may have happened at μs time scale and therefore it is difficult to perform the measurements.
In addition, the protein is undergoing a folding upon self-assembling and hence a blue shift in the maximum intensity is expected. Therefore, I believe it would be helpful to have full fluorescence spectra at least at the initial and end time points to assess whether or not there is a shift in the fluorescence or rather only an intensity increase. In case there is no shift, this could explain that these Trp are exposed, in the monomer and the assembled forms.
- Label bars in the EM images.
- Could you explain why the heat curves from the ITC measurements are not analyzed and fitted? The fitted curves can provide useful information on enthalpy, entropy and stoichiometry.
- Legend Fig. 5, line 394: remove “in different concentrations of C124 as indicated”.
- It seems from the ANS binding measurements in Fig. 2A that the protein is either a PMG or a MG. That could be determined by plotting the CD values according to Uversky, V. N. Protein Sci. 2002.
- In the discussion part, line 518, the claim is that the protein undergoes folding from unstructured to NLPs that are rich in β-sheet structures. This happens in the presence of nucleic acids. It would be interesting to see whether this is the case at pH12. CD spectra of the protein either in the presence of nucleic acids or at pH12 could be performed and compare the NLP structures as these might give hints on whether the mechanism of self-assembly is the same as suggested in the manuscript.
- I believe that the discussion about the specificity towards nuclei acids should be toned down especially that the full-length protein and/or other co-factors, i.e. cellular chaperones, might be involved in determining the specificity of binding to specific nuclei acids and they are not considered in the present study. Therefore, I suggest to turn the conclusion into a possible speculation rather a solid statement.
English improvement, especially in the discussion:
. line 528-529 should be reformulated
. line 531: factor “leading” to the “protein” assembly
. line 550: better “understood”

Reviewer 2 ·

Basic reporting

The authors carried out an expanded biophysical characterization of the C-terminal truncated form of the hepatitis C virus core protein. In my opinion, the results provide a significant amount of new structural information on this protein that may guide further studies on the relationship between its structure and biological function. As a minor note, I found a few sentences a bit difficult to understand. There are a few minor typographical or grammatical errors (e.g., NPLs, line 63; ...whereas that the..., line 130).

Experimental design

The experiments are relevant and appear to have been competently done.

Validity of the findings

The major conclusions are reasonably supported by the results presented. The findings provide some new structural information on this protein that may help to explain, to a certain extent, some aspects related to its function in the infectious cycle.

Additional comments

No comments.

---

## Round 0.2 · accepted · Accept

I really appreciate your efforts dedicated to addressing all the critical points of the reviewers.